# Increasing human motor skill acquisition by driving theta–gamma coupling

Haya Akkad[1,2]*, Joshua Dupont-Hadwen[1], Edward Kane[1], Carys Evans[1], Liam Barrett[3], Amba Frese[3], Irena Tetkovic[3], Sven Bestmann[1,4]*†, Charlotte J Stagg[2,5,6]*†

[1]Department for Clinical and Movement Neuroscience, UCL Queen Square Institute of Neurology, University College London, London, United Kingdom; [2]Wellcome Centre for Integrative Neuroimaging, FMRIB, Nuffield Department of Clinical Neurosciences, University of Oxford, Oxford, United Kingdom; [3]Department of Experimental Psychology, University College London, London, United Kingdom; [4]Wellcome Centre for Human Neuroimaging, UCL Queen Square Institute of Neurology, University College London, London, United Kingdom; [5]Oxford Centre for Human Brain Activity, Wellcome Centre for Integrative Neuroimaging, Department of Psychiatry, University of Oxford, Oxford, United Kingdom; [6]MRC Brain Network Dynamics Unit, University of Oxford, Oxford, United Kingdom

**Abstract** Skill learning is a fundamental adaptive process, but the mechanisms remain poorly understood. Some learning paradigms, particularly in the memory domain, are closely associated with gamma activity that is amplitude modulated by the phase of underlying theta activity, but whether such nested activity patterns also underpin skill learning is unknown. Here, we addressed this question by using transcranial alternating current stimulation (tACS) over sensorimotor cortex to modulate theta–gamma activity during motor skill acquisition, as an exemplar of a non-hippocampal-dependent task. We demonstrated, and then replicated, a significant improvement in skill acquisition with theta–gamma tACS, which outlasted the stimulation by an hour. Our results suggest that theta–gamma activity may be a common mechanism for learning across the brain and provides a putative novel intervention for optimizing functional improvements in response to training or therapy.

*For correspondence:
haya.akkad.14@ucl.ac.uk (HA);
s.bestmann@ucl.ac.uk (SB);
charlotte.stagg@ndcn.ox.ac.uk
(CJS)

†These authors contributed
equally to this work

Competing interest: The authors
declare that no competing
interests exist.

Reviewing Editor: Thorsten
Kahnt, Northwestern University,
United States

## Editor's evaluation

This study provides evidence that increasing theta-gamma phase-amplitude coupling, which is thought to be critical for hippocampal memory, can improve non-hippocampal motor learning. This conclusion is based on two experiments showing that transcranial alternating current stimulation over M1 improves motor learning relative to sham stimulation and an active control condition. The findings will be interesting to neuroscientists and clinicians, as they elucidate mechanisms of motor learning and have implications for improving outcomes for patients recovering from motor impairments.

## Introduction

The acquisition of motor skills is a central part of our everyday lives, from learning new behaviours such as riding a bike to the recovery of function after brain injury such as a stroke (*Yarrow et al., 2009*; *Krakauer et al., 2019*; *Dhawale et al., 2017*; *Diedrichsen and Kornysheva, 2015*). Better understanding of the mechanisms underpinning skill acquisition, to develop mechanistically informed

strategies and tools to promote skill learning in healthy and pathological movement, is therefore a high-priority scientific and clinical goal.

Acquisition of motor skills is linked to a number of cortical and subcortical brain regions, but among these, primary motor cortex (M1) is thought to play a central role (*Yarrow et al., 2009*; *Krakauer et al., 2019*; *Diedrichsen and Kornysheva, 2015*; *Sanes and Donoghue, 2000*), making this a key target for neurorehabilitative interventions (*Ward et al., 2019*; *Kang et al., 2016*; *Allman et al., 2016*). However, the neurophysiological changes through which one might be able to promote skill acquisition in M1 are poorly understood, substantially hampering the development of novel interventions.

Outside the motor domain, the mechanisms underpinning learning have been extensively studied in the hippocampus, where theta-amplitude-coupled mid-gamma frequency activity (θ–γ phase-amplitude coupling [PAC]) has been hypothesized as a key learning-related mechanism. A prominent feature of hippocampal theta (4–8 Hz) activity is its co-incidence with higher-frequency activity in the γ range (30–140 Hz). Gamma coherence in the hippocampus alters during learning (*Montgomery and Buzsáki, 2007*) and memory retrieval (*Yamamoto et al., 2014*), and its relative synchrony during task predicts subsequent recall (*Fell et al., 2001*; *Headley and Weinberger, 2011*).

Hippocampal activity at different gamma frequencies is coupled to distinct phases of the underlying theta rhythm, suggesting that the precise relationship between gamma activity and theta phase may be important for function (*Bragin et al., 1995*; *Lasztóczi and Klausberger, 2014*; *Colgin, 2015*). For example, 60–80 Hz activity, which increases significantly during memory encoding, is coupled to the peak of the underlying theta oscillation (*Lopes-Dos-Santos et al., 2018*).

θ–γ PAC appears to be a conserved phenomenon across the cortex, and has been hypothesized as a fundamental operation of cortical computation in neocortical areas (*Fries, 2009*). For example, In the sensory cortices, it provides a neural correlate for perceptual binding (*Lisman and Jensen, 2013*). In the pre-frontal cortex, externally driven θ–γ PAC directly influences spatial working memory performance and global neocortical connectivity when gamma oscillations are delivered coinciding with the peak, but not the trough of theta waves (*Muellbacher et al., 2001*). It is proposed that the theta rhythm forms a temporal structure that organizes gamma-encoded units into preferred phases of the theta cycle, allowing careful processing and transmission of neural computations (*Watrous, 2015*). In the motor cortex, gamma oscillations at approximately 75 Hz are observed during movement (*Crone et al., 1998*; *Pfurtscheller and Lopes da Silva, 1999*; *Pfurtscheller et al., 2003*; *Muthukumaraswamy, 2011*; *Crone et al., 2006*; *Muthukumaraswamy, 2010*; *Nowak et al., 2017*), and an increased 75 Hz activity has been observed in dyskinesia, suggesting a direct pro-kinetic role (*Swann et al., 2016*; *Swann et al., 2018*). As in the hippocampus, M1 gamma oscillations are modulated by theta activity, with 75 Hz activity in human M1 being phase locked to the peak of the theta waveform (*Canolty et al., 2006*).

However, whether theta–gamma coupling plays a similar role in non-hippocampal-dependent skill learning in neocortical regions as it does in the hippocampus has not yet been determined. We therefore wished to test the hypothesis that θ–γ PAC is a conserved mechanism for learning across the brain, and therefore may provide a target for influencing the acquisition of new behaviour. To investigate the functional role of θ–γ PAC in learning outside the hippocampus, we modulated local theta–gamma activity via externally applied alternating current stimulation (tACS), a non-invasive form of brain stimulation that can interact with and modulate neural oscillatory activity in the human brain in a frequency-specific manner (*Ali et al., 2013*; *Feurra et al., 2011*; *Zaehle et al., 2010*), over M1 during learning of an M1-dependent ballistic thumb abduction task (*Dupont-Hadwen et al., 2019*) learning outside the hippocampus, we modulated local theta–gamma activity via externally applied alternating current stimulation (tACS), a non-invasive form of brain stimulation that can interact with and modulate neural oscillatory activity in the human brain in a frequency-specific manner (*Ali et al., 2013*; *Feurra et al., 2011*; *Zaehle et al., 2010*), over M1 during learning of an M1-dependent ballistic thumb abduction task (*Dupont-Hadwen et al., 2019*). We chose this task because it shows robust behavioural improvement in a relatively short period of time and performance improvement is underpinned by plastic changes in M1 (*Classen et al., 1998*; *Muellbacher et al., 2001*; *Muellbacher et al., 2002*). This encoding of kinematic details of the practiced movement is commonly regarded as a first step in skill acquisition (*Classen et al., 1998*).

We reasoned that if θ–γ PAC is a key mechanism for motor skill learning then interacting with θ–γ PAC, specifically with 75 Hz gamma activity applied at the theta peak (*Lopes-Dos-Santos et al.,*

*2018*; *Canolty et al., 2006*), via tACS should have the capacity to modulate skill acquisition in healthy human participants, putatively via a change in local excitability. Moreover, if the functional role of this theta–gamma PAC is indeed critically dependent on the gamma activity occurring at a specific phase of theta activity then any behavioural effect should be specific to the theta phase at which the gamma was applied. To address this question, we therefore derived a waveform with gamma applied during the trough of the theta activity as an active control.

We first conducted an exploratory single-blinded experiment, in which we tested for the influence of theta–gamma-coupled stimulation on skill acquisition. This experiment revealed that when applied externally over right M1, gamma coupled to the peak of a theta envelope (TGP) substantially enhanced motor skill acquisition, compared to sham and an active stimulation control. Based on these results, we conducted a second, double-blind, pre-registered, sham-controlled experiment, which confirmed the beneficial effect of TGP on motor skill acquisition.

## Results

One hundred and four healthy participants performed a M1-dependent ballistic thumb abduction task with their left hand (*Dupont-Hadwen et al., 2019*; *Rogasch et al., 2009*; *Rosenkranz et al., 2007*) while tACS was applied over the right M1. Volunteers trained to increase thumb abduction acceleration in their left, non-dominant, thumb over 5 blocks of 70 trials each (*Figure 1*).

Fifty-eight participants (age: 24 ± 5.1 years, 37 females) participated in experiment 1, and were randomly assigned to one of three experimental groups, which received either 20 min of tACS over right primary motor cortex, or sham. Similar to a previous study in the spatial working memory domain (*Alekseichuk et al., 2016*), in the active tACS condition, participants received (1) theta–gamma peak (TGP) stimulation (*Figure 1A*), whereby gamma frequency (75 Hz) stimulation was delivered during the peak of a 6 Hz theta envelope as is found naturally in the human motor cortex (*Canolty et al., 2006*), or (2) an active control, theta–gamma trough (TGT) stimulation, whereby the gamma stimulation was delivered in the negative half of the theta envelope. For sham stimulation, 6 Hz theta was briefly ramped up for 10 s, and then ramped down again. Participants performed the skill learning task during the stimulation, and for approximately 15 min after cessation of stimulation.

### TGP stimulation improves motor skill acquisition

We first wished to assess whether participant performance improved with training, regardless of stimulation. As expected, skill increased in all three groups over the course of the experiment (repeated measures analysis of variance [ANOVA] with one factor of block [1–6] and one factor of condition [TGP, TGT, and sham], main effect of block $F_{(2.203, 121.187)}$ = 85.122, p < 0.001). However, the stimulation groups differed significantly in their skill acquisition (main effect of condition $F_{(2, 55)}$ = 3.396, p = 0.041; condition × block interaction $F_{(4.407, 121.187)}$ = 2.692, p = 0.03). Post hoc tests (using Tukey correction for multiple comparisons) revealed a significant difference between TGP and sham (p = 0.04, 95% CI [0.50, 21.78]) and no significant difference between TGT and sham (p = 0.766, 95% CI [−7.52, 13.79]) or TGP and TGT (p = 0.162, 95% CI [−2.40, 18.35]). To further explore the interaction effect, we ran an analysis of simple effects to determine the effect of the condition factor (TGP, TGT, and sham) at each level of the block factor (1–6). This revealed a significant simple effect of condition during stimulation blocks $F_{(2, 55)}$ = 4.13, p = 0.021. In line with our primary hypothesis, follow-up analyses demonstrated a 26% larger acceleration gain from baseline during TGP stimulation, compared with sham condition (independent t-test $t_{(36)}$ = 3.052, p = 0.004, Cohen's d = 0.98, *Figure 2A*).

There were no significant differences in baseline performance between TGP, TGT, and sham conditions as demonstrated by a simple effects analysis of the factor of Condition (TGP, TGT, and sham) at the level of baseline block 1 $F_{(2, 55)}$ = 0.30, p = 0.743.

This first experiment established the relevant role of theta–gamma-coupled tACS over M1 on motor skill learning in healthy participants, here expressed through an increase in learning. This effect was most effective when gamma frequency stimulation was coupled to the peak of the underlying theta frequency stimulation waveform, as opposed to when it was coupled to the trough of theta. We next sought to confirm this result in an independent cohort, and to further assess the duration of this improvement post-stimulation.

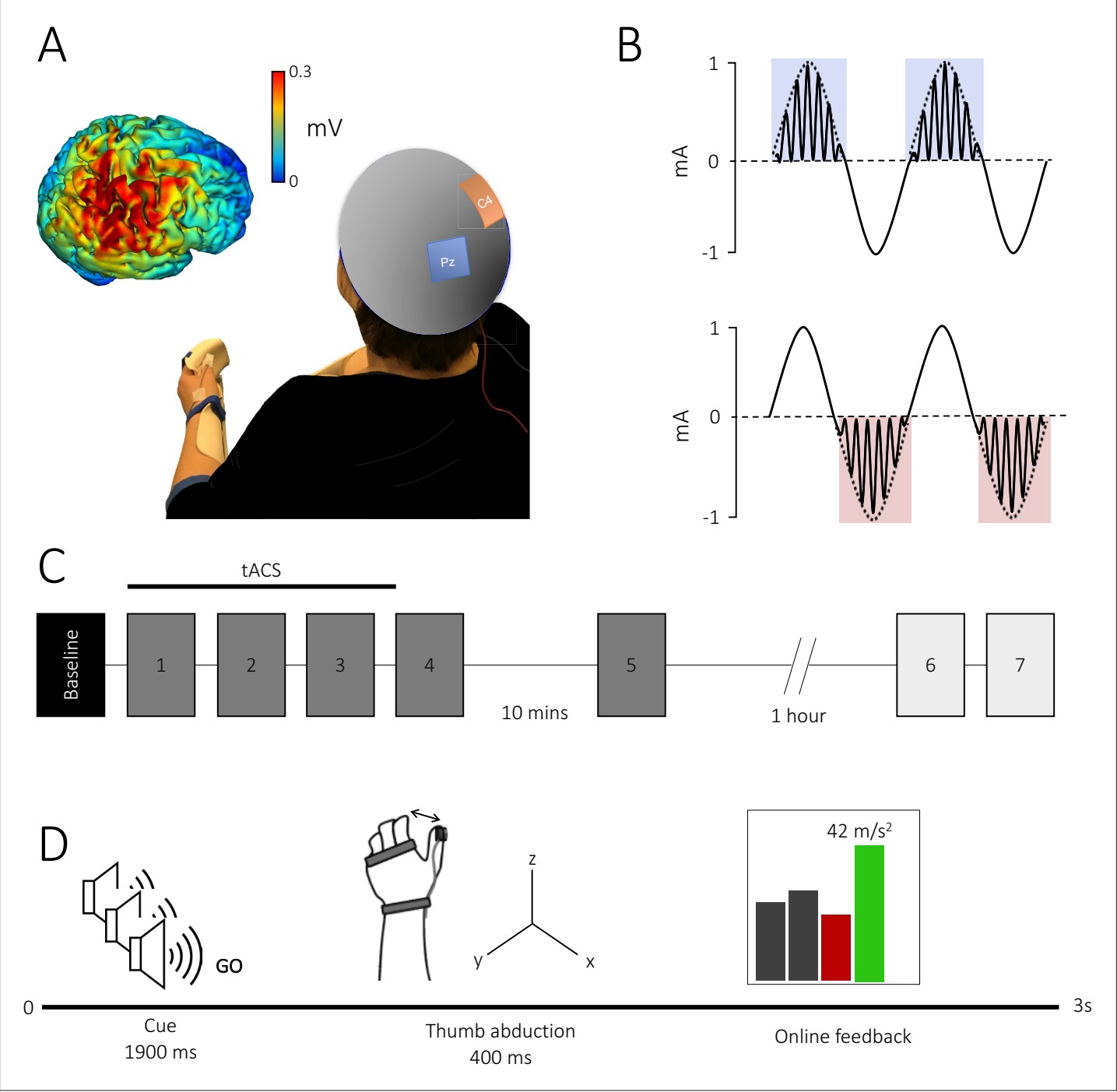

**Figure 1.** Theta–gamma transcranial alternating current stimulation (tACS) protocol and task. (A) Electrode montage: the theta–gamma tACS montage was delivered with one electrode centred over right M1 (red, C4) and the other over the parietal vertex (blue, Pz). Insert: electrical field distribution projected on a rendered reconstruction of the cortical surface in a single individual, demonstrating significant current within M1. (B) tACS waveform: a 75 Hz gamma rhythm was amplitude modulated by the peak (theta–gamma peak [TGP]; upper panel) or trough (theta–gamma trough [TGT]; lower panel) envelope of a 2 mA peak-to-peak 6 Hz theta rhythm. (C) Experimental design: all subjects performed a baseline block, followed by five task blocks. In experiment 2, to assess the duration of behavioural effects, subjects performed an additional two task blocks 75 min after the initial task was complete. Each block consisted of 70 trials with an inter-block interval of 2 min, apart from a 10 min and 1 hr break after blocks 4 and 5, respectively. Stimulation was delivered for 20 min during the first three blocks. (D) Trial design: all trials began with three auditory warning tones acting as a ready-steady-go cue. At the third tone, participants abducted their thumb along the x-axis as quickly as possible and were given online visual feedback of their performance via a screen positioned in front of them. Feedback was presented as a scrolling bar chart with the magnitude of acceleration displayed on a trial-by-trial basis; a green bar indicated acceleration was higher than the previous trial and a red bar indicated the opposite.

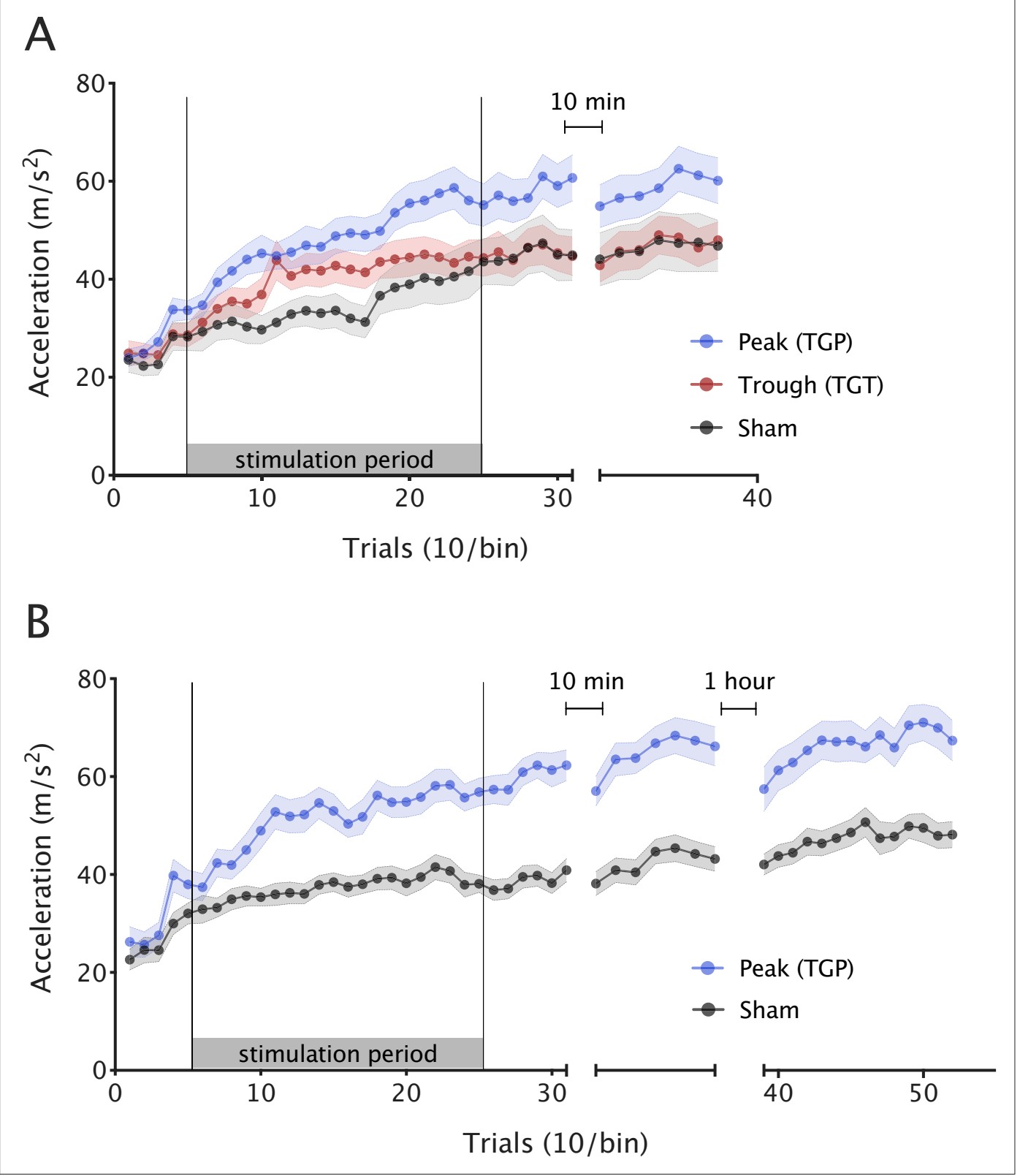

**Figure 2.** Theta–gamma peak (TGP)-transcranial alternating current stimulation (tACS) enhances motor skill acquisition. Mean ballistic thumb abduction acceleration for each stimulation condition. Each point represents the mean of 10 trials across participants and the error bars depict the standard error between participants. (A) Experiment 1: during stimulation, TGP significantly increased skill acquisition over the course of the experiment (i.e. acceleration gain), compared to sham and theta–gamma trough (TGT). (B) Experiment 2: when replicated in an independent sample, skill acquisition

*Figure 2 continued on next page*

*Figure 2 continued*

was again significantly greater in the TGP stimulation group compared with sham. This effect was maintained for at least 75 min after stimulation.

The online version of this article includes the following figure supplement(s) for figure 2:

**Figure supplement 1.** Transcranial alternating current stimulation (tACS) does not modulate behavioural variability.

## Behavioural effects of TGP stimulation are replicable

In order to try to replicate our results from experiment 1, we conducted a double-blind, pre-registered (https://osf.io/xjpef) replication experiment in an independent sample of 46 participants (age 24 ± 4.1, 32 females, all right handed). Because our first experiment had shown the largest effect on skill learning with TGP stimulation, we now focussed on this condition. Participants were randomised to either TGP stimulation or sham. The experimental protocol was identical to experiment 1, except that we additionally included a probe to test retention at 1 hr after the end of stimulation. There was no significant difference in baseline performance between TGP and sham conditions ($t(44) = 0.734$, p = 0.467).

As in experiment 1, participants in both conditions showed an improvement in performance throughout the experiment (repeated measures ANOVA, one factor of block [1–8], one factor of condition [TGP and sham]; main effect of block $F(3.302,145.239) = 72.912$, p < 0.001; *Figure 2B*).

However, there was a significant difference in skill acquisition between the two conditions (main effect of condition ($F(1,44) = 27.241$, p < 0.001); block × condition interaction $F(3.302,145.239) = 7.258$, p < 0.001). The TGP group achieved significantly greater acceleration gain compared to sham during stimulation ($t(44) = 4.201$, p < 0.001, Cohen's $d = 1.24$).

Bang's blinding index (BI; *Bang et al., 2004*) indicated successful blinding in both real and sham stimulation groups. Blinding indices were 0.07 and −0.03 in the TGP and sham groups, respectively.

## Motor skill gains are retained post-stimulation

We next wished to explore whether the behavioural effects of stimulation outlasted the stimulation period, or whether skill in this group returned to baseline after stimulation had ceased. Comparing the two groups at 75 min post-stimulation demonstrated that the TGP group had a significantly faster acceleration than the sham group ($t(44) = 3.430$, p = 0.001, Cohen's $d = 1.01$).

## tACS does not significantly modulate the variability or latency of responses

tACS may increase skill acquisition by changing one or more different aspects of behaviour. Non-invasive brain stimulation approaches have previously been demonstrated to increase behavioural variability in tasks similar to that implemented here (*Teo et al., 2010*). First, we investigated whether tACS significantly modulated the variability in the maximum acceleration achieved. We ran ANOVAs on the coefficient of variation for each subject for each block, with a within-subject factor of block and between-subject factor of condition for each experiment separately. This revealed a main effect of block in both experiments (E1: $F(5,275)=17.1$, p < 0.001; E2: $F(7,308) = 13.8$, p < 0.001), reflecting a general decrease in variability during the task, but no main effect of condition (E1: $F(2,55) = 2.36$, p = 0.104; E2: $F(1,44) = 0.14$, p = 0.90) and no block × condition interaction (E1: $F(10,275) = 1.15$, p = 0.329; E2: $F(7,308) = 1.60$, p = 0.14; *Figure 2—figure supplement 1*).

Second, we wished to investigate whether tACS-modulated response time. We therefore ran an ANOVA with a within-subject factor of block and between-subject factor of condition. In experiment 1, there was a significant main effect of block ($F(2.18,120) = 11.68$, p < 0.001) and condition ($F(2,55) = 4.66$, p = 0.013), but no significant block × condition interaction ($F(4,63,120) = 0.195$, p = 0.95). However, when we repeated this analysis for experiment 2 there was no significant effect of block ($F(1,44) = 0.014$, p = 0.905), and no significant block by condition interaction ($F(3.60,158.45) = 0.372$, p = 0.30).

## Discussion

Theta-amplitude-modulated gamma activity may provide an important mechanism for non-hippocampal-dependent skill acquisition. We used non-invasive brain stimulation to modulate θ–γ

PAC in human primary motor cortex in two separate cohorts, one a pre-registered, double-blind study and demonstrated that externally applied θ–γ PAC during a motor task increases skill acquisition in healthy adults. This behavioural improvement was critically dependent on the phase relationship of the theta and gamma components of the stimulation.

## Behavioural improvements depend on the phase of theta–gamma coupling

Our results suggest that driving γ activity during the peak, but not the trough, of θ oscillations improves motor skill acquisition. θ–γ PAC has consistently been demonstrated to relate to learning in the rodent CA1 (*Bragin et al., 1995*; *Lasztóczi and Klausberger, 2014*; *Colgin, 2015*; *Lopes-Dos-Santos et al., 2018*), where oscillations in the θ (5–12 Hz) band become dominant during active exploration (*O'Keefe and Recce, 1993*), and have been widely hypothesized to allow information coming into CA1 from distant regions to be divided into discrete units for processing (*Buzsáki, 2002*; *Buzsáki and Moser, 2013*). A prominent feature of hippocampal theta activity is its co-incidence with higher-frequency activity in the γ range (30–140 Hz). Gamma coherence in the hippocampus alters during learning (*Montgomery and Buzsáki, 2007*) and memory retrieval (*Yamamoto et al., 2014*), and its relative synchrony during task predicts subsequent recall (*Fell et al., 2001*; *Headley and Weinberger, 2011*). Non-invasively stimulating the human temporal cortex during memory encoding using tACS to increase θ–γ coupling has been variously shown to impair (*Lara et al., 2018*) or strengthen (*Reinhart and Nguyen, 2019*) hippocampal memory formation.

Hippocampal activity at different frequencies within the gamma band is coupled to distinct phases of the underlying theta rhythm, suggesting that the precise relationship between gamma activity and theta phase may be important for function (*Bragin et al., 1995*; *Lasztóczi and Klausberger, 2014*; *Colgin, 2015*). For example, 60–80 Hz activity, which increases significantly during memory encoding, is coupled to the peak of the underlying theta oscillation (*Lopes-Dos-Santos et al., 2018*).

θ–γ PAC appears to be a conserved phenomenon across the cortex and has been hypothesized as a fundamental operation of cortical computation in neocortical areas (*Fries, 2009*; *Lisman and Jensen, 2013*; *Alekseichuk et al., 2016*). Supporting this hypothesis, a recent human study demonstrated an improvement in working memory using tACS (*Reinhart and Nguyen, 2019*). However, no study to date has shown that θ–γ PAC can modulate non-hippocampal-dependent *learning* as we do here.

## 75 Hz activity has a pro-kinetic role in M1 and relates to skill acquisition

Our experiments indicate that 75 Hz activity, coupled to 6 Hz oscillations, can improve motor skill acquisition. We chose 75 Hz stimulation for two reasons: it is implicated in learning in the hippocampus and physiologically, M1 gamma activity centred around 75 Hz occurs at the peak of ongoing theta activity (*Canolty et al., 2006*) and is ubiquitous in studies of human movement. 75 Hz activity only occurs during actual, rather than imagined, movement (*Muthukumaraswamy, 2010*), and shows topographical specificity within M1 (*Crone et al., 2006*). Its hypothesized pro-kinetic role is further supported by the finding of a pathological increase in narrow-band 75 Hz activity within M1 in hyper-kinetic patients with Parkinson's disease (*Swann et al., 2016*). Our group have previously shown that the degree of response to 75 Hz tACS predicts subsequent learning potential, further highlighting a role for 75 Hz activity not only in movement but in skill acquisition (*Nowak et al., 2017*). Here, we demonstrate that a more physiological approach to delivering gamma stimulation by coupling it to theta rhythms leads to theta-phase-specific improvements in skill acquisition – something that may allow the development of more targeted therapeutic interventions.

## Behavioural benefits of theta–gamma PAC may be mediated by decreases in inhibition

Decreases in M1 GABAergic activity are a central mechanism for motor plasticity (*Stagg et al., 2009*; *Clarkson et al., 2010*; *Stagg et al., 2011*; *Blicher et al., 2015*; *Bachtiar et al., 2015*; *Traub et al., 1996*). However, it is not yet clear *how* these decreases alter behaviour. θ–γ PAC may be a candidate mechanism for this: M1 gamma activity arises from GABAergic inter-neuronal micro-circuits involving layer V Parvalbumin+ ve neurons (*Whittington and Traub, 2003*; *Bartos et al., 2007*; *Cabral et al., 2011*; *Chen et al., 2017*; *Sohal et al., 2009*; *Ni et al., 2016*; *Whittington et al., 2011*; *Masamizu*

*et al., 2014*) thought to be involved in motor learning (*Johnson et al., 2017*). In slice preparations, theta–gamma coupling within M1 arises spontaneously from layer V when GABA activity is blocked (*Johnson et al., 2020*). In humans, modulating M1 75 Hz activity in humans using tACS leads to a decrease in local GABAergic activity, the magnitude of which predicts motor learning ability on a subject-by-subject basis (*Nowak et al., 2017*). The effects of low-frequency tACS may be mediated through cyclically inducing a phase of enhanced excitation (peak) followed by a phase of reduced excitation (trough). If decreases in M1 GABAergic activity is necessary for motor plasticity (*Bang et al., 2004*; *Teo et al., 2010*; *O'Keefe and Recce, 1993*; *Buzsáki, 2002*; *Buzsáki and Moser, 2013*), then phases of enhanced excitation (or reduced inhibition) would offer an optimal entrainment window for excitatory rhythms, such as pro-kinetic 75 Hz gamma.

Given the extensive evidence for decreases in GABAergic activity for motor cortical plasticity, it may be that gamma activity, particularly synchronization of gamma activity via theta oscillations, represents an emergent signature of learning that might be targeted to improve behaviour, though the cellular and layer specificity of our findings remain to be determined.

## The behavioural effects of tACS are not driven by changes in variability or latency of responses

There are a number of potential mechanisms by which the behavioural improvements we observed might have arisen. Previous studies have demonstrated that skill improvement due to non-invasive brain stimulation might occur via an increase in the variability of behavioural responses (*Teo et al., 2010*), but this does not seem to be the case here. Additionally, it is possible that our measure of skill learning was confounded by a stimulation-induced change in response time, but the data do not support this hypothesis. However, given the strongly pro-kinetic role of 75 Hz activity in M1, further studies should look at the specific components of motor behaviour this tACS protocol may modulate to identify the precise aspects of motor skill acquisition theta–gamma tACS may modulate.

## Anatomical- and frequency specificity of behavioural effects

θ–γ PAC has been suggested as a mechanism by which anatomically distant brain regions become functionally connected (*Fries, 2009*). We deliberately chose a task that is M1 dependent, thereby providing us with a cortical target for our stimulation, and have not set out to target more than one node in the network. We are confident that we are actively stimulating M1: our tACS protocol induces excitability changes in M1, suggesting a significant physiological effect in this region, and the electrical field simulation demonstrates a significant current within M1 due to tACS. However, this does not rule out that the behavioural effects we observe arise from multiple nodes, and that there is a contribution of the parietal electrode: indeed as with all tACS studies, the current is relatively widespread across the cortex. This hypothesis remains to be tested.

Here, we tested an a priori hypothesis about theta–gamma PAC, and its role in non-hippocampal-dependent skill acquisition. We did not test other frequency couplings, and so we cannot claim that similar effects would not be seen with other cross-frequency stimulation paradigms. Similarly, we did not directly test whether θ–γ PAC was superior to either θ or γ stimulation alone. However, by using an active TGT control condition, which delivered the same θ and γ stimulation, and only varied the phase of the θ at which the γ was present, if the behavioural effects seen were solely dependent on either frequency alone then both the peak and trough conditions would have improved learning, which was not the case. Previous studies have shown that gamma stimulation alone can improve learning (*Asamoah et al., 2019*; *Moisa et al., 2016*), but not to such as degree as θ–γ PAC (*Alekseichuk et al., 2016*).

## Lack of TGT behavioural effect supports the hypothesis that tACS directly modulates neural activity

There has been some recent controversy about the contribution of direct stimulation of the underlying neural tissue versus other mechanisms (*Krause et al., 2019*; *Vieira et al., 2020*; *Vöröslakos et al., 2018* ) to the behavioural and physiological effects of tACS, although recent work strongly supports the argument that tACS directly entrains ongoing neural activity (*Krause et al., 2019*; *Vieira et al., 2020*). While this paper does not aim to directly address this question, we are confident that our behavioural effects result from direct effects of the current in the brain. Firstly, tACS at the current

densities used here have been demonstrated to entrain single-neuron activity in non-human primates (*Krause et al., 2019*), suggesting at least that direct neuronal entrainment is a possible mechanism. Secondly, although stimulation of peripheral scalp nerves has recently been suggested as a putative explanation for behavioural effects of tACS (*Asamoah et al., 2019*), in experiment 1, we used an inverted waveform as an active control to rule out effects driven by peripheral stimulation, and in experiment 2, there was successful blinding to stimulation type. Collectively, this suggests that the sensory sensations that may arise from stimulation did not substantially differ between active and sham conditions.

## Conclusions

In conclusion, we wished to test whether theta–gamma PAC was an important mechanism in non-hippocampally dependent learning in humans. Using a novel non-invasive brain stimulation approach in humans that emulates known neurophysiological activity patterns during learning (*Lopes-Dos-Santos et al., 2018*; *Lisman and Jensen, 2013*; *Canolty et al., 2006*), we demonstrated, and then replicated, a substantial behavioural improvement due to stimulation. While the neural underpinnings of this functional outcome need to be explored, this result offers a new technique not only to understand physiological mechanisms of human neuroplasticity, but also potentially a putative novel adjunct therapy for promoting post-stroke recovery.

## Materials and methods

### Experiment 1

Fifty-eight participants (24 ± 5.1 years, 37 females) gave their written informed consent to participate in the experiments in accordance with local ethics committee approval. Participants were right handed and had no contraindications for tACS. Participants were randomly assigned to one of three tACS conditions (*N* = 20 per condition): (1) TGP stimulation (*Figure 1A*), whereby gamma frequency (75 Hz) stimulation was delivered during the peak of a 6 Hz theta envelope, (2) an active control: TGT stimulation where the gamma stimulation was delivered in the negative half of the theta envelope, and (3) sham stimulation. Participants were blinded to the type of stimulation delivered and naive to the purpose of the experiment.

### Experimental setup

Participants performed a ballistic thumb abduction training task requiring abduction of their left (non-dominant) thumb with maximal acceleration (*Dupont-Hadwen et al., 2019*; *Rogasch et al., 2009*; *Rosenkranz et al., 2007*). Participants were seated with their left arm slightly abducted, with the elbow flexed to 45° (where 0° is full extension) and the forearm semi-pronated with the palm facing inwards. The left hand was chosen to avoid ceiling effects that might be present in the dominant hand. The arm, wrist, and proximal interphalangeal joints were secured in a plastic custom-built arm fixture to prevent the unintentional contribution of whole hand movement to the ballistic acceleration, though the thumb was left free to move (*Figure 1C*).

The acceleration of the thumb was measured in the *x*-axis (abduction plane) using an accelerometer (ACL300; Biometrics Ltd, UK) attached to the distal phalanx of the thumb. Recording from the accelerometer was confined to one axis to allow for good skill improvement by providing simplified feedback for the participant (*Dupont-Hadwen et al., 2019*; *Rogasch et al., 2009*; *Rosenkranz et al., 2007*).

### Behavioural task

Participants performed ballistic thumb abduction movements of their left hand at a rate of 0.4 Hz indicated by a ready-steady-go procedure, with each of three auditory tones (400 Hz, 300 ms duration) spaced at 500 ms intervals. Participants were instructed to move their thumbs at the onset of the third auditory tone. The behavioural task was separated into six blocks (*Figure 1B*). Participants performed an initial baseline block of 30 trials. This was followed by four blocks separated by a break of at least 2 min to minimize fatigue, and a final block separated by a 10 min break. Each of these five blocks consisted of 70 trials with a 30 s break between every 35 trials to avoid within block fatigue. Participants were asked to remain at rest during breaks, avoiding any thumb movement.

In all blocks except the baseline block, participants were instructed to move as fast as possible and were encouraged to try to increase their acceleration on every trial. Participants were given visual feedback about the acceleration of their movements on a trial-by-trial basis (*Figure 1C*). Feedback was presented as a scrolling bar chart with the magnitude of the current acceleration plotted after each trial. If the acceleration on the current trial was greater than on the previous trial, the bar was plotted in green, and if it was less the bar was plotted in red. If a movement was made too early or too late (i.e. movement outside a 300 ms window centred on one second after the first tone), no acceleration feedback was given; instead, the message 'too early' or 'too late' was presented. Additionally, participants were informed of their progress by displaying a moving average of acceleration values over the preceding 10 trials, indicated by a line plotted on the screen over the locations of the 10 consequential trials.

In the baseline block, participants were told to move as closely as possible to the onset of the third tone, and feedback about the temporal accuracy of the movement was given by the experimenter.

## Behavioural data analysis

Data were analysed via Matlab (Mathworks). The maximal acceleration was calculated for each trial, and any trials with a maximum acceleration less than 4.9 $ms^2$ were rejected (*Dupont-Hadwen et al., 2019*). Additionally, if movements were made too early or too late, that is the onset of acceleration of the movement lay more than 300 ms before or after the expected movement time, they were also rejected (*Dupont-Hadwen et al., 2019*). Together, this approach led to 1.45 ± 0.94 (mean ± standard deviation [SD]) trials being removed per block of 70 trials in experiment 1, and 0.88 ± 0.99 (mean ± SD) trials removed per block of 70 trials in experiment 2. There was no statistical difference between the number of trials being removed per block in each condition (experiment 1: mixed ANOVA, block × condition [$F(5.409,148.742) = 1.649$, $p = 0.145$]; experiment 2: mixed ANOVA, block × condition [$F(2.8,137.4) = 1.05$, $p = 0.396$]).

## Transcranial alternating current stimulation

tACS was delivered via a DC stimulator in AC mode (NeuroConn DC-Stimulator Plus) through a pair of sponge surface electrodes (5 × 5 $cm^2$). Saline was used as a conducting medium between the scalp and the electrodes. The anode was centred over the right primary motor cortex (C4) and the cathode was positioned over the parietal vertex (Pz), in accordance with the international 10–20 EEG system. Impedance was kept below 10 kΩ. The electrode positions were based on simulation of current flow across the brain, using HD-Explore software (Soterix Medical Inc, New York) which uses a finite-element-method approach to model electrical field intensities throughout the brain (*Datta et al., 2013*). This confirmed that current was directed to the primary motor cortex (*Figure 1A*).

The TGP condition consisted of 20 min continuous, sinusoidal 6 Hz (theta) stimulation at an intensity of 2 mA peak-to-peak, coupled with bursts of a sinusoidal 75 Hz (gamma) rhythm amplitude modulated by the positive theta phase (0–180°; *Figure 1A*). The TGT condition consisted of 20 min continuous, sinusoidal 6 Hz (theta) stimulation at an intensity of 2 mA peak-to-peak, coupled with bursts of a sinusoidal 75 Hz (gamma) rhythm amplitude modulated by the negative theta phase (180–360°). Finally, the sham condition consisted of a 10 s continuous sinusoidal 6 Hz stimulation.

The theta–gamma waveforms were custom coded on the Matlab software and delivered to the NeuroConn stimulator via a data acquisition device (National Instruments USB-6259 BNC). Theta–gamma stimulation was then delivered to the scalp surface electrodes through the NeuroConn stimulator in 'remote' mode. Sham stimulation was delivered directly through the NeuroConn stimulator. tACS was administered in a between-subject design. Participants were randomized to receive either 10 s of sham stimulation during the first training block or 20 min of TGP or TGT stimulation during the first three training blocks. Participants were blinded to the stimulation condition used and naive to the purpose of the experiment.

## Statistical analyses

Data were tested for normality using the Kolmogorov–Smirnov test. Statistical analyses were performed using SPSS. We used a two-way mixed ANOVA with two independent variables, 'condition' (between-subject variable) and 'block' (within-subject variable). Acceleration in $ms^2$ was our only dependent variable. Where there was a significant block × condition interaction, we analysed the simple effect of

condition within levels of block. Post hoc *t*-tests were conducted as appropriate and multiple comparisons were corrected for using the Tukey HSD test. When sphericity assumptions were violated, results are reported with a Greenhouse–Geisser correction.

## Experiment 2

Forty-four participants (age 24 ± 4.1 years, 32 females) gave their written informed consent to participate in the experiments in accordance with local ethics committee approval. Participants were right handed and had no contraindications for tACS. We performed a pre-registered, double-blinded replication of experiment 1 (TGP and sham only) in an independent sample. The experimental design was pre-registered in full on the Open Science Framework (https://osf.io/xjpef). The experimental design was identical to experiment 1, except in the following aspects.

### Power Calculation

Sample size was calculated based on the Cohen's *d* effect size of the mean improvement in performance from baseline between the TGP and sham conditions in experiment 1. Given a Cohen's $d = 0.98$, $1 - \beta = 0.95$ and $\alpha = 0.05$ this gave a sample size of 24 per group (G*Power), and allowing for a 10% loss of data, we recruited 27 participants per condition.

### Blinding

On the day of testing, a researcher not involved in data analysis and blinded to experimental protocol and rationale (AF, IT, and LB) collected the data and interacted with the participant. Another researcher (HA), not involved in data collection and blinded during data analysis, setup the stimulation condition on the day of testing, but did not interact with the participant. Unblinding was performed following the completion of data collection and analysis. Participants were naive to the purpose of the experiment.

Participants completed a blinding questionnaire at the end of the experiment that required them to identify whether they believed they had received real or sham stimulation. To assess the effectiveness of our blinding, we used Bang's BI, where a BI of 1 suggests complete unblinding, a BI of 0 random guessing and a BI of −1 opposite guessing.

### Behavioural task

The behavioural task parameters were identical to those in experiment 1, but now with an additional two training blocks separated from the previous six blocks by a break of 1 hr (*Figure 1B*). During the 1 hr break, participants remained seated and at rest while watching a documentary (Planet Earth, season 1 episode 10). The additional seventh and eighth blocks were separated by a break of at least 2 min to minimize fatigue and each consisted of 70 trials, with a 30 s break between every 35 trials to minimize within block fatigue. Participants were asked to remain at rest during breaks, avoiding any thumb movement.

### Transcranial alternating current stimulation

Stimulation parameters were identical to those in experiment 1, but only included the TGP and sham conditions. Both the participant and the experimenter were blinded to the stimulation condition used.

## Acknowledgements

HA holds a doctoral fellowship funded by Brain Research UK (552175). SB was funded by Brain Research UK (201617-03) and Dunhill Medical Trust (RPGF1810/93). CJS holds a Sir Henry Dale Fellowship, funded by the Wellcome Trust and the Royal Society (102584/Z/13/Z). The work was supported by the NIHR Biomedical Research Centre, Oxford and the NIHR Oxford Health Biomedical Research Centre. The Wellcome Centre for Integrative Neuroimaging is supported by core funding from the Wellcome Trust (203139/Z/16/Z). The Wellcome Centre for Human Neuroimaging is supported by funding from the Wellcome Trust (203147/Z/16/Z).

## Additional information

### Funding

| Funder | Grant reference number | Author |
|--------|------------------------|--------|
| Royal Society | Sir Henry Dale Fellowship 102584/Z/13/Z | Charlotte J Stagg |
| Brain Research UK | 201617-03 | Sven Bestmann |
| Brain Research UK | Graduate Student Fellowship | Haya Akkad |
| Wellcome Trust | Sir Henry Dale Fellowship - 102584/Z/13/Z | Charlotte J Stagg |

The funders had no role in study design, data collection, and interpretation, or the decision to submit the work for publication.

### Author contributions

Haya Akkad, Data curation, Formal analysis, Investigation, Methodology, Project administration, Validation, Visualization, Writing - original draft, Writing – review and editing; Joshua Dupont-Hadwen, Methodology; Edward Kane, Carys Evans, Liam Barrett, Data curation; Amba Frese, Sven Bestmann, Conceptualization, Data curation, Funding acquisition, Investigation, Methodology, Resources, Supervision, Validation, Writing – review and editing; Irena Tetkovic, Charlotte J Stagg, Conceptualization, Data curation, Funding acquisition, Investigation, Methodology, Resources, Supervision, Validation, Visualization, Writing – review and editing

### Author ORCIDs

Haya Akkad ![ORCID] http://orcid.org/0000-0002-5621-3318
Sven Bestmann ![ORCID] http://orcid.org/0000-0002-6867-9545

### Ethics

Human subjects: Ethical permission for this study was granted by the University College London Research Ethics Committee (UCLREC: 6285/001). Written informed consent was obtained from all volunteers prior to data collection.

### Decision letter and Author response

Decision letter https://doi.org/10.7554/eLife.67355.sa1
Author response https://doi.org/10.7554/eLife.67355.sa2

## Additional files

### Supplementary files
• Transparent reporting form

### Data availability

All data generated or analysed during this study are included in the manuscript and freely available on the open science framework (https://osf.io/xjpef). Details of data analysis, experimental design and protocol were pre-registered prior to data collection and freely available on the open science framework - Registration form: osf.io/xjpef; Files: osf.io/452f8/files/.

The following previously published datasets were used:

| Author(s) | Year | Dataset title | Dataset URL | Database and Identifier |
|-----------|------|---------------|-------------|-------------------------|
| Akkad H, Dupont-Hadwen J, Bestmann S, Stagg CJ | 2018 | Improving motor learning via phase-amplitude coupled theta-gamma tACS | https://osf.io/xjpef | Open Science Framework, 10.17605/OSF.IO/XJPEF |

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
