## [Editor Report]

This study provides evidence that increasing theta-gamma phase-amplitude coupling, which is thought to be critical for hippocampal memory, can improve non-hippocampal motor learning. This conclusion is based on two experiments showing that transcranial alternating current stimulation over M1 improves motor learning relative to sham stimulation and an active control condition. The findings will be interesting to neuroscientists and clinicians, as they elucidate mechanisms of motor learning and have implications for improving outcomes for patients recovering from motor impairments.

---

## [Decision Letter]

**Decision letter after peer review:**

Thank you for submitting your article "Increasing human motor skill acquisition by driving theta-gamma coupling" for consideration by *eLife*. Your article has been reviewed by 3 peer reviewers, one of whom is a member of our Board of Reviewing Editors, and the evaluation has been overseen by Richard Ivry as the Senior Editor. The following individual involved in review of your submission has agreed to reveal their identity: Michael A Nitsche (Reviewer #3).

Essential revisions:

1. Reviewers agreed that the TMS experiment (Figure 3) does not provide sufficient evidence to draw strong conclusions about the mechanisms by which theta-gamma tACS enhances motor skill acquisition. Reviewers agreed that the sample (N=10) is too small, the effect too weak, and the interaction effect difficult to interpret. Our preference would be to have you repeat this experiment in a larger sample. However, we also felt that there was enough meat in the paper that this should not be required as part of revision. If you are in a position to run a better powered TMS experiment, great-we certainly think it would be a nice addition to the paper. Otherwise, we ask that you discuss this limitation and tone down the conclusions about mechanisms in a revision.

2. Reviewers would also like to see additional analyses of these TMS data. Specifically, please combine blocks 1-3 into "stimulation blocks" and 4-7 as "non-stimulation blocks" and run the ANOVA this way. A significant interaction effect should be followed up by post-hoc tests, comparing TGP vs sham and TGP vs TGT in stimulation and non-stimulation blocks.

*Reviewer #1:*

This manuscript reports the results from two between-subject experiments testing the role of theta-gamma phase amplitude coupling (PAC) for non-hippocampal motor skill learning. Both experiments use transcranial alternating current stimulation (tACS) over right M1. Experiment 1 (single-blind, N=58) tests three groups with theta-gamma stimulation, where gamma stimulation is delivered either at the peak (TGP) or the through of the theta envelope (TGT), or sham. The results show that TGP relative to sham and TGT increases the acceleration of left thumb abduction. This basic effect is replicated in a second (preregistered, double-blind) experiment (N=46), which further shows that these effects last for at least 1 hour after stimulation offset. Additional analyses show that results are not driven by effects on variability or response times. These experiments are rigorous and the results provide convincing support for the idea that theta-gamma PAC during the peak of the theta rhythm facilitates non-hippocampal skill acquisition.

An additional within-subject experiment (N=10) tests whether this effect is driven by increased cortical excitability. This is assessed using motor evoked potentials (MEP), measured using TMS. The authors report a significant stimulation by block interaction, but no main effect of stimulation. These MEP effects are less convincing for several reasons. First, the experiment is potentially underpowered and the effects are weak. Second, the stimulation/condition by time interaction is difficult to interpret without a main effect of condition at the time of stimulation.

In summary, the study is rigorous and the results are novel and important. They convincingly show that theta-gamma tACS improves non-hippocampal skill learning. These findings are important in two ways. First, they show that theta-gamma PAC plays a role beyond hippocampal-dependent learning. Second, they may have implications for therapy and treatment. However, the results from the TMS experiment are less convincing, leaving it open whether increased cortical excitability is the mechanism by which theta-gamma tACS facilitates motor skill acquisition.

Comments for the authors:

1. My only major concern is related to the TMS experiment. The interaction effect is not very robust and I am not sure the interaction is even the right effect to look at. The interaction appears to be driven just as much by a higher MEP amplitude in the TGP condition during stimulation that decreases afterwards, as it is driven by a low MEP amplitude in the sham condition that increases after stimulation offset. Is there even a difference between conditions during the stimulation period? Also with N=10 this experiment is likely underpowered. I don't think these results are conclusive enough to inform the mechanism by which the stimulation works. My recommendation is to either collect a larger sample or to remove these results and pertaining discussions entirely.

2. Experiment 1: The authors show a difference between TGP and sham, but is there also a difference between TGP and TGT?

3. Is the p value in line 116 is a typo? Should it be p=0.041 instead of p=0.41?

*Reviewer #2:*

Scientifically, this is a very nice piece of work. The active control is well-chosen to emphasize the specific importance of the theta phase of stimulation. The pre-registered replication improves confidence in the reliability of the effect. Finally, showing this stimulation increases cortical excitability offers a potential mechanism underlying these effects.

I believe the authors could do a better job framing the paper, for which I have some suggestions below. I also point out several small issues with the writing, which should be fixed upon any revision.

In general, the discussion of the hippocampus is too prominent. For instance, it is mentioned in the second sentence of the abstract, and it is a bit jarring coming to the third sentence and subsequently finishing the abstract to realize that the paper has nothing to do with the hippocampus but rather leans on that literature for motivation (which is fine!). For instance, the authors could change the abstract to the following: "Some learning paradigms are closely associated with gamma activity that is amplitude-modulated by the phase of underlying theta activity, but whether such nested activity patterns also underpin skill acquisition is unknown." I also believe the emphasis from lines 51-63 should be on learning and theta-gamma coupling first, with references to research on the hippocampus coming as an ancillary point rather than centering the hippocampus. Such a shift would be subtle, but I believe it would be effective here. Additionally, some aspects of the framing, such as mentioning subregion CA1, are too granular for the motor study here.

Figure 3: What do the authors make of the reduction in the TGP condition after the stimulation? Is this significant on its own? If so, the authors should speculate about whether this might reflect some sort of "fatigue" or "refractory" effect following the enhanced stimulation.

It seems appropriate to describe something about the motor task used in the Introduction. I am familiar with many motor tasks, but not this one (until this paper), so the authors may improve the paper by describing why this one was used here (rather than others).

*Reviewer #3:*

In this contribution, the authors explored the importance of theta-gamma coupling, an oscillatory brain activity pattern shown to be relevant for hippocampus-dependent learning, for motor skill acquisition, which does critically involve neocortical areas, but not the hippocampus, and thus the general importance of this oscillatory brain activity for learning and memory formation, in healthy humans. The main results of the study show that non-invasive brain stimulation with transcranial alternating currents (tACS) over the sensorimotor cortex indeed improved motor skill learning, and this effect lasted relevantly beyond stimulation. This thus establishes a new mechanistic foundation for learning and memory formation in humans, which might have future applications in rehabilitation. One relevant limitation of the study is that the respective theta gamma protocol also enhanced motor cortex excitability, and the experiment does not allow to conclude if the effects depend critically on the excitability enhancement, which might also be induced by other stimulation protocols not involving theta gamma coupling.

Strengths

1. Innovative concept exploring the relevance of specific brain oscillation patterns for learning and memory formation beyond hippocampal learning.

2. Causal test of the relevance of theta-gamma coupling for motor skill acquisition by non-invasive brain stimulation.

3. The introduction gives a comprehensive overview about the contribution of theta-gamma activity to various cognitive and behavioral functions.

4. The experimental design does involve not only behavioral tests, but also physiological exploration of tACS-induced excitability alterations via TMS-generated MEPs, which helps to clarify physiological mechanisms further.

5. The experimental design explores the specificity of the effects by coupling theta and gamma activity in different ways.

6. The results of the main study were replicated in a second experiment, which increases reliability.

7. Important control measures were conducted for variability, and latency of responses, which helps to clarify specificity of the results.

8. A mechanistic explanation for the effect of theta gamma coupling on motor skill acquisition in given.

9. In the discussion, limitations of the study are clearly mentioned.

Weaknesses

1. In the introduction, the contribution of theta gamma coupling to working memory performance is not mentioned, which is a pity, because here theta gamma tACS over prefrontal areas was shown to alter performance. These results show thus the functional relevance of these oscillations in neocortical areas for cognitive functions.

2. The non-specialized reader is not introduced into mechanisms of action of tACS.

3. It does not become clear if successful blinding was tested in experiment 1.

4. For the TMS measures, the statistics do not allow to identify the specific timing-dependency of differences between the TGP and sham conditions.

5. It would have been good to obtain physiological and behavioral measures in the same group of subjects. This would have allowed to correlate results, and thus make stronger conclusions about the interdependency of excitability, and behavioral alterations.

6. Since only the TGP protocol improved learning, but this protocol also enhanced excitability, it cannot be excluded that not the oscillatory entrainment, but the excitability alteration caused behavioral improvement, which might mean that the oscillations themselves are not critical for the effects, as far as I can see.

7. In the discussion, it does not become quite clear why gamma activity specifically at the theta peak, but not at the trough, should reduce GABA activity.

8. For application of tACS, by the experimental design it cannot be completely excluded that the Pz electrode contributed to the effects, although this is unlikely.

Comments for the authors:

1. It might be good to add information in the introduction about the relevance of theta gamma coupling, also in connection with tACS, for working memory performance in humans.

2. It might be good to add a para to the introduction which explains tACS for non-specialized readers.

3. Please add the blinding index, if available, also for experiment 1.

4. For the TMS measures, it might make sense to add post hoc tests comparing at least TGP, and sham conditions to explore the timing-dependency of the effects.

5. The authors might want to add information, why gamma activity specifically at the theta peak, but not at the trough, should reduce GABA activity, if such information is available.

6. The authors might want to discuss a possible contribution of the parietal electrode to the effects, or exclude this by a control experiment.

7. The authors might want to discuss if it could be the case that not the oscillations, but the excitability enhancement could have been critical for the induced effects. One option to test this would have been a control condition with other stimulation protocols not including oscillations tested here.

[Editors' note: further revisions were suggested prior to acceptance, as described below.]

Thank you for resubmitting your work entitled "Increasing human motor skill acquisition by driving theta-gamma coupling" for further consideration by *eLife*. Your revised article has been reviewed by 3 peer reviewers, one of whom is a member of our Board of Reviewing Editors, and the evaluation has been overseen by Richard Ivry as the Senior Editor.

The manuscript has been improved but there are some remaining issues that need to be addressed, as outlined below:

All reviewers agreed that you have adequately addressed several of the essential issues. However, there were lingering concerns regarding the results from the TMS experiment and whether they provide evidence for potential mechanisms of tACS.

Specifically, the additional analyses reveal that there was no significant effect of stimulation condition during the stimulation period and that the condition x block interaction was driven by an effect of TGT vs. sham after the stimulation. Furthermore, there was no evidence for differences between TGP and either sham or TGT, either during or after the stimulation. It is unclear whether this is a true null result or whether there are no effects because of insufficient power. In any case, given this experiment does not provide clear evidence for an effect of the critical tACS protocol (i.e., TGP), reviewers agreed that these data do not help to inform potential mechanisms of tACS.

Unless the situation in your lab has changed such that you are now able to collect additional TMS data to run a well-powered TMS experiment, we'd ask you to remove the TMS experiment and any reference to it from the manuscript.

*Reviewer #1:*

The authors have been responsive to several of my initial concerns, but my major point about the TMS experiment was not sufficiently addressed.

The authors re-analyzed these data and found no effect of TGP or TGT during the stimulation, but reduced MEPs after TGT stimulation compared to sham and TGP. Importantly, there was no difference between TGP and sham during or after the stimulation. It is therefore unclear why the authors conclude that they found "a pattern of enhanced cortical excitability during TGP stimulation compared to sham and TGT stimulation." As far as I can see, there was no evidence for such a pattern during (not reported) or after the stimulation (p>0.14). So don't think the author's conclusion that "these TMS data provide valuable initial support for phase-entrainment as a potential mechanism of action for tACS" is supported by the data.

*Reviewer #2:*

I commend the authors for performing a nice revision and again for submitting an interesting paper. They have done an especially good job at this stage by offering more measured interpretations. I recommend publication.

PS. There are still a few instances of beginning sentences with numerals. Please fix these.

*Reviewer #3:*

All major issues were solved appropriately.

---

## [Author Response]

Essential revisions:1. Reviewers agreed that the TMS experiment (Figure 3) does not provide sufficient evidence to draw strong conclusions about the mechanisms by which theta-gamma tACS enhances motor skill acquisition. Reviewers agreed that the sample (N=10) is too small, the effect too weak, and the interaction effect difficult to interpret. Our preference would be to have you repeat this experiment in a larger sample. However, we also felt that there was enough meat in the paper that this should not be required as part of revision. If you are in a position to run a better powered TMS experiment, great-we certainly think it would be a nice addition to the paper. Otherwise, we ask that you discuss this limitation and tone down the conclusions about mechanisms in a revision.

We thank the editor and reviewers for allowing us to review and clarify our TMS findings. We would agree with the reviewers that the results of the TMS study are limited by the small sample size and therefore provide inconclusive evidence regarding mechanism of action of tACS. Unfortunately, in the current circumstances we are unable to collect more TMS data and are grateful for the editor’s understanding of this. However, despite the small sample, we strongly believe that these findings provide valuable preliminary evidence that is worth highlighting for future studies.

In line with reviewer’s suggestions, we have made substantial changes to the manuscript to fully acknowledge this limitation and the preliminary nature of the TMS findings (see reviewer responses below for more detail). We have toned down the conclusions about the mechanisms in the revision. We have additionally run new statistical analyses of the TMS data, which we hope the reviewers and editor agree have made the results clearer to interpret (see the responses to individual reviewers below for more detail).

2. Reviewers would also like to see additional analyses of these TMS data. Specifically, please combine blocks 1-3 into "stimulation blocks" and 4-7 as "non-stimulation blocks" and run the ANOVA this way. A significant interaction effect should be followed up by post-hoc tests, comparing TGP vs sham and TGP vs TGT in stimulation and non-stimulation blocks.

We thank the reviewers for this recommendation, which helps clarify the TMS data. In the revised manuscript, we now combine the data into ‘stimulation blocks’ and ‘post-stimulation blocks’ and run a RM ANOVA followed by post-hoc t-tests examining the interaction effect (see reviewer responses below for more detail).

Reviewer #1:This manuscript reports the results from two between-subject experiments testing the role of theta-gamma phase amplitude coupling (PAC) for non-hippocampal motor skill learning. Both experiments use transcranial alternating current stimulation (tACS) over right M1. Experiment 1 (single-blind, N=58) tests three groups with theta-gamma stimulation, where gamma stimulation is delivered either at the peak (TGP) or the through of the theta envelope (TGT), or sham. The results show that TGP relative to sham and TGT increases the acceleration of left thumb abduction. This basic effect is replicated in a second (preregistered, double-blind) experiment (N=46), which further shows that these effects last for at least 1 hour after stimulation offset. Additional analyses show that results are not driven by effects on variability or response times. These experiments are rigorous and the results provide convincing support for the idea that theta-gamma PAC during the peak of the theta rhythm facilitates non-hippocampal skill acquisition.An additional within-subject experiment (N=10) tests whether this effect is driven by increased cortical excitability. This is assessed using motor evoked potentials (MEP), measured using TMS. The authors report a significant stimulation by block interaction, but no main effect of stimulation. These MEP effects are less convincing for several reasons. First, the experiment is potentially underpowered and the effects are weak. Second, the stimulation/condition by time interaction is difficult to interpret without a main effect of condition at the time of stimulation.In summary, the study is rigorous and the results are novel and important. They convincingly show that theta-gamma tACS improves non-hippocampal skill learning. These findings are important in two ways. First, they show that theta-gamma PAC plays a role beyond hippocampal-dependent learning. Second, they may have implications for therapy and treatment. However, the results from the TMS experiment are less convincing, leaving it open whether increased cortical excitability is the mechanism by which theta-gamma tACS facilitates motor skill acquisition.

Thank you. We are pleased to see that the reviewer appreciates the important implications of theta-gamma coupling in non-hippocampal skill learning. In what follows, we address the reviewer’s concerns, particularly with regards to the TMS experiment where we have made substantial effort to clarify findings and address limitations.

Comments for the authors:1. My only major concern is related to the TMS experiment. The interaction effect is not very robust and I am not sure the interaction is even the right effect to look at. The interaction appears to be driven just as much by a higher MEP amplitude in the TGP condition during stimulation that decreases afterwards, as it is driven by a low MEP amplitude in the sham condition that increases after stimulation offset. Is there even a difference between conditions during the stimulation period? Also with N=10 this experiment is likely underpowered. I don't think these results are conclusive enough to inform the mechanism by which the stimulation works. My recommendation is to either collect a larger sample or to remove these results and pertaining discussions entirely.

We thank the reviewer for this comment, which was highlighted in the editor’s summary above. Unfortunately given the current circumstances we are not able to acquire more TMS data. We agree that in this limited sample size, the mechanism of action of theta-gamma tACS remains inconclusive. However, we strongly believe that these preliminary findings offer valuable insight into a potential mechanism of action that compliments recent findings from single-unit recordings in non-human primates that is worth exploring in future studies. We have substantially revised our manuscript to clearly address the limitations of our TMS findings. In addition we have run new analyses to clarify the interaction effect presented. We hope that these revisions clarify the findings from, and limitations of, the TMS data, and present it in an appropriate context.

As per reviewers’ recommendations, we combined the TMS data into ‘stimulation blocks’ and ‘post-stimulation blocks’ and ran a RM ANOVA followed by post-hoc t-tests examining the interaction effect i.e. the timing-dependency of the effect. These changes are included in the revised Results section as follows (lines 206-216):

‘MEPs were normalised to baseline and we performed a repeated-measures ANOVA, with one factor of condition (TGP, TGT and Sham) and one factor of block (stimulation blocks and post-stimulation). […] However, MEPs were significantly reduced after TGP stimulation compared with during TGP stimulation (*t(9)=2.861, p=0.019*), which might reflect a refractory period following enhanced cortical excitability, though this remains speculative.’

We further interpret these TMS results and address the limitation of our small sample size in the revised discussion (Lines 261-271):

‘To explore changes in cortical activity in response to tACS, we quantified changes in TMS-evoked MEPs during theta-gamma peak, trough and sham stimulation. […] However, these TMS data provide valuable initial support for phase-entrainment as a potential mechanism of action for tACS and more generally, for TMS-tACS protocols as an effective tool to explore the phase-specific pattern of neural responses in the human cortex.’

2. Experiment 1: The authors show a difference between TGP and sham, but is there also a difference between TGP and TGT?

We apologise for not including this. There was no significant difference between TGP and TGT. We have now clarified the Results section as follows (lines 140-144):

‘Post-hoc tests (using Tukey correction for multiple comparisons) revealed a significant difference between TGP and sham (p=0.04) and no significant difference between TGT and sham (p=0.766) or TGP and TGT (p=0.162). To further explore the interaction effect, we ran an analysis of simple effects to determine the effect of Condition (TGP, TGT, sham) at each level of Block (1-6).’

3. Is the p value in line 116 is a typo? Should it be p=0.041 instead of p=0.41?

We thank the Reviewer for pointing out this typo. We have now corrected this in the Results section (lines 139):

‘…Effect of Condition *F(2, 55)=3.396 , p=0.41 p=0.041…’*

Reviewer #2:Scientifically, this is a very nice piece of work. The active control is well-chosen to emphasize the specific importance of the theta phase of stimulation. The pre-registered replication improves confidence in the reliability of the effect. Finally, showing this stimulation increases cortical excitability offers a potential mechanism underlying these effects.I believe the authors could do a better job framing the paper, for which I have some suggestions below. I also point out several small issues with the writing, which should be fixed upon any revision.In general, the discussion of the hippocampus is too prominent. For instance, it is mentioned in the second sentence of the abstract, and it is a bit jarring coming to the third sentence and subsequently finishing the abstract to realize that the paper has nothing to do with the hippocampus but rather leans on that literature for motivation (which is fine!). For instance, the authors could change the abstract to the following: "Some learning paradigms are closely associated with gamma activity that is amplitude-modulated by the phase of underlying theta activity, but whether such nested activity patterns also underpin skill acquisition is unknown." I also believe the emphasis from lines 51-63 should be on learning and theta-gamma coupling first, with references to research on the hippocampus coming as an ancillary point rather than centering the hippocampus. Such a shift would be subtle, but I believe it would be effective here. Additionally, some aspects of the framing, such as mentioning subregion CA1, are too granular for the motor study here.

We thank the reviewer for giving us the opportunity to re-frame the manuscript and clarify the motivation of the work. We have revised our discussion of the hippocampus in the abstract and introduction. We hope this helps guide the reader more clearly through the work.

We have made the following changes to the abstract (lines 31-36):

‘Hippocampal learning is closely associated with gamma activity, which is amplitude-modulated by the phase of underlying theta activity. […] Some learning paradigms, particularly in the memory domain, are closely associated with gamma activity that is amplitude-modulated by the phase of underlying theta activity, but whether such nested activity patterns also underpin skill learning is unknown.’

In the introduction, we added the following to emphasise the importance of theta-gamma coupling outside the hippocampus (lines 69-74):

‘In the pre-frontal cortex, externally-driven θ-γ PAC directly influences spatial working memory performance and global neocortical connectivity when gamma oscillations are delivered coinciding with the peak, but not the trough of theta waves [35]. It is proposed that the theta rhythm forms a temporal structure that organizes gamma-encoded units into preferred phases of the theta cycle, allowing careful processing and transmission of neural computations [Watrous et al., 2015].’

We have removed mention of subregion CA1 (lines 56-59):

‘In rodent hippocampal area CA1, oscillations in the θ (5-12 Hz 4-8 Hz) band become dominant during active exploration [9], and have been hypothesised to allow information coming into CA1 from distant regions to be divided into discrete units for processing [10,11].’

With regards to the order, we believe the introduction flows most logically from the hippocampus first – where the fundamentals of cross-frequency coupling and its role in learning were first established – followed by more recent work identifying cross-frequency coupling in the neocortex (although no work has been performed to date on the role of theta-gamma coupling in skill learning in these regions). We believe that this framing helps establish a clear motivation to explore whether theta-gamma coupling in the neocortex plays a similar role in non-hippocampal learning as it does in the hippocampus. We hope the changes in the amended manuscript have helped clarify the motivation of the work, and have removed the concentration on the hippocampus, which we agree did not aid the relevant framing of the paper.

Figure 3: What do the authors make of the reduction in the TGP condition after the stimulation? Is this significant on its own? If so, the authors should speculate about whether this might reflect some sort of "fatigue" or "refractory" effect following the enhanced stimulation.

We thank the reviewer for making this observation. We have added the following to address this point (lines 214-216):

‘However, MEPs were significantly reduced after TGP stimulation compared with during TGP stimulation (*t(9)=2.861, p=0.019*), which might reflect a refractory period following enhanced cortical excitability, though this remains speculative.’

It seems appropriate to describe something about the motor task used in the Introduction. I am familiar with many motor tasks, but not this one (until this paper), so the authors may improve the paper by describing why this one was used here (rather than others).

We thank the reviewer for allowing us to clarify this important point. We have added the following to the introduction (lines 88-92):

‘We chose this task because it shows robust behavioural improvement in a relatively short period of time and performance improvement is underpinned by plastic changes in M1 [Classen et al., 1998; Muellbacher et al. 2001; Muellbacher et al. 2002]. This encoding of kinematic details of the practiced movement is commonly regarded as a first step in skill acquisition [Classen et al., 1998].’

Reviewer #3:[…] 1. It might be good to add information in the introduction about the relevance of theta gamma coupling, also in connection with tACS, for working memory performance in humans.

We have added the following to address this point (lines 69-74):

‘In the pre-frontal cortex, externally-driven θ-γ PAC directly influences spatial working memory performance and global neocortical connectivity when gamma oscillations are delivered coinciding with the peak, but not the trough of theta waves [35]. It is proposed that the theta rhythm forms a temporal structure that organizes gamma-encoded units into preferred phases of the theta cycle, allowing careful processing and transmission of neural computations [Watrous et al., 2015].’

2. It might be good to add a para to the introduction which explains tACS for non-specialized readers.

Thank you for this important point – we have added the following to the manuscript (lines 83-87):

‘…we modulated local theta-gamma activity via externally applied alternating current stimulation (tACS), a non-invasive form of brain stimulation that can interact with and modulate neural oscillatory activity in the human brain in a frequency-specific manner [Ali et al., 2013; Feurra et al., 2011; Zaehle et al., 2010], over M1 during learning of an M1-dependent ballistic thumb abduction task skill [32].’

3. Please add the blinding index, if available, also for experiment 1.

Unfortunately, a blinding index is not available for experiment 1 as blinding data was not collected.

4. For the TMS measures, it might make sense to add post hoc tests comparing at least TGP, and sham conditions to explore the timing-dependency of the effects.

We thank the reviewer for this suggestion, which has helped clarify the TMS results. We have made substantial changes to the analyses of the TMS data. As per reviewers’ recommendations, we combined the data into ‘stimulation blocks’ and ‘post-stimulation blocks’ and ran a RM ANOVA followed by post-hoc t-tests examining the interaction effect. These changes are revised in the Results section as follows (lines 206-216):

‘MEPs were normalised to baseline and we performed a repeated-measures ANOVA, with one factor of condition (TGP, TGT and Sham) and one factor of block (stimulation blocks and post-stimulation). […] However, MEPs were significantly reduced after TGP stimulation compared with during TGP stimulation (*t(9)=2.861, p=0.019*), which might reflect a refractory period following enhanced cortical excitability, though this remains speculative.’

5. The authors might want to add information, why gamma activity specifically at the theta peak, but not at the trough, should reduce GABA activity, if such information is available.

We have added the following to address this point (lines 279-285):

‘The effects of low frequency tACS may be mediated through cyclically inducing a phase of enhanced excitation (peak) followed by a phase of reduced excitation (trough). […] This hypothesis would be in line with the increase in cortical excitability we observed during TGP, but not TGT stimulation, and the tACS peak phase-preference demonstrated in single-unit recording studies [Johnson et al., 2020]’

6. The authors might want to discuss a possible contribution of the parietal electrode to the effects, or exclude this by a control experiment.

Thank you for this important point. We agree with the reviewer that there might be a possible contribution of the parietal electrode. However, we deliberately chose an M1-dependent task to minimise the potential confounds. We have amended the discussion to include a greater comment on the potential contribution of other nodes (lines 304-309):

‘We are confident that we are actively stimulating M1: our tACS protocol induces excitability changes in M1, suggesting a significant physiological effect in this region, and the electrical field simulation demonstrates a significant current within M1 due to tACS. […] This hypothesis remains to be tested.’

7. The authors might want to discuss if it could be the case that not the oscillations, but the excitability enhancement could have been critical for the induced effects. One option to test this would have been a control condition with other stimulation protocols not including oscillations tested here.

We thank the reviewer for allowing us to expand on this important point. It would seem parsimonious to suggest an explanation where the oscillations led to an increase in excitability, which led to the behavioural effects. Indeed, it seems likely that this is the mechanism by which the oscillations exert their behavioural effects. We have added the following to the Discussion section to clarify this line of argument (lines: 279-285):

‘The effects of low frequency tACS may be mediated through cyclically inducing a phase of enhanced excitation (peak) followed by a phase of reduced excitation (trough). […] This hypothesis would be in line with the increase in cortical excitability we observed during TGP, but not TGT stimulation, and the tACS peak phase-preference demonstrated in single-unit recording studies [Johnson et al., 2020]’

References:

Ali MM, Sellers KK and Fröhlich F (2013). Transcranial alternating current stimulation modulates large-scale cortical network activity by network resonance. Journal of Neuroscience 33(27), 11262-11275.

Classen J, Liepert J, Wise SP, Hallett M and Cohen LG (1998). Rapid plasticity of human cortical movement representation induced by practice. Journal of Neurophysiology 79(2):1117-23.

Feurra M, Paulus W, Walsh V and Kanai R (2011). Frequency specific modulation of human somatosensory cortex. Frontiers in Psychology 2, 13.

Johnson L, Aleksiechuck I, Krieg J, et al. (2020). Dose-dependent effects of transcranial alternating current stimulation on spike timing in awake nonhuman primates. Science Advances 6: eaaz2747.

Muellbacher W, Ziemann U, Boroojerdi B, Cohen L and Hallett M (2001). Role of the human motor cortex in rapid motor learning. Experimental Brain Research 136(4): 431-8.

Muellbacher W, Ziemann U, Wissel J, Dang N, Kofler M, Facchini S, Boroojerdi B, Poewe W and Hallett M (2002). Early consolidation in human primary motor cortex. Nature 415(6872): 640-4.

Watrous AJ, Deuker L, Fell J and Axmacher N (2015). Phase-amplitude coupling supports phase coding in human ECoG. *eLife*, 4 e07886

Zaehle T, Rach S, and Herrmann CS (2010). Transcranial alternating current stimulation enhances individual α activity in human EEG. PLoS One, 5(11), e13766.

[Editors' note: further revisions were suggested prior to acceptance, as described below.]

The manuscript has been improved but there are some remaining issues that need to be addressed, as outlined below:All reviewers agreed that you have adequately addressed several of the essential issues. However, there were lingering concerns regarding the results from the TMS experiment and whether they provide evidence for potential mechanisms of tACS.Specifically, the additional analyses reveal that there was no significant effect of stimulation condition during the stimulation period and that the condition x block interaction was driven by an effect of TGT vs. sham after the stimulation. Furthermore, there was no evidence for differences between TGP and either sham or TGT, either during or after the stimulation. It is unclear whether this is a true null result or whether there are no effects because of insufficient power. In any case, given this experiment does not provide clear evidence for an effect of the critical tACS protocol (i.e., TGP), reviewers agreed that these data do not help to inform potential mechanisms of tACS.Unless the situation in your lab has changed such that you are now able to collect additional TMS data to run a well-powered TMS experiment, we'd ask you to remove the TMS experiment and any reference to it from the manuscript.

We are very pleased that the Reviewers felt that we had addressed the substantial majority of their comments. In response to the main comment remaining from Reviewer 1, and highlighted by the Editor, we have removed all mention of the TMS experiment from the manuscript. We have responded to all the Reviewers’ comments point-by-point below.

Reviewer #1:The authors have been responsive to several of my initial concerns, but my major point about the TMS experiment was not sufficiently addressed.The authors re-analyzed these data and found no effect of TGP or TGT during the stimulation, but reduced MEPs after TGT stimulation compared to sham and TGP. Importantly, there was no difference between TGP and sham during or after the stimulation. It is therefore unclear why the authors conclude that they found "a pattern of enhanced cortical excitability during TGP stimulation compared to sham and TGT stimulation." As far as I can see, there was no evidence for such a pattern during (not reported) or after the stimulation (p>0.14). So don't think the author's conclusion that "these TMS data provide valuable initial support for phase-entrainment as a potential mechanism of action for tACS" is supported by the data.

We understand the Reviewer’s point and have removed the TMS experiment from the manuscript as the Editor requested.

Reviewer #2:I commend the authors for performing a nice revision and again for submitting an interesting paper. They have done an especially good job at this stage by offering more measured interpretations. I recommend publication.PS. There are still a few instances of beginning sentences with numerals. Please fix these.

We apologise that we had missed these: the remaining instances were all in the TMS sections and have therefore now been removed.